# PROMPT-BASED LENGTH CONTROLLED GENERATION WITH REINFORCEMENT LEARNING

## ABSTRACT

Large language models (LLMs) like ChatGPT and GPT-4 have attracted great attention given their surprising performance on a wide range of NLP tasks. Length controlled generation of LLMs emerges as an important topic, which enables users to fully leverage the capability of LLMs in more real-world scenarios like generating a proper answer or essay of a desired length. In addition, the autoregressive generation in LLMs is extremely time-consuming, while the ability of controlling this generated length can reduce the inference cost by limiting the length. Therefore, we propose a prompt-based length control method to achieve high-accuracy length controlled generation. In particular, we adopt reinforcement learning with the reward signal given by either trainable or rule-based reward models, which further enhances the length-control ability of LLMs by rewarding outputs that follows pre-defined control instruction. To enable rule-based inference, we also introduce standard prompt extractor to collect the standard control information from users' input. Experiments show that our method significantly improves the accuracy of prompt-based length control for summarization task on popular datasets like CN-NDM and NYT. Both the standard prompt extractor and the RL-tuned model have show strong generalization ability to unseen control prompt templates.

## 1 INTRODUCTION

For recent popular GPT-style LLMs like ChatGPT and GPT-4 (Radford et al., 2018; 2019; Liu et al., 2023b; OpenAI, 2023), various studies have been conducted on them, and the inference efficiency and computational cost often draw concerns from the community (Zhang et al., 2023; Zhao et al., 2023; Bubeck et al., 2023). Since its generation is in an autoregressive manner, the inference cost increases continually with the growing of decoding steps. Meanwhile, users of LLMs usually have an expected length of generated texts, no matter for writing an essay or summary, knowledge QA or dialogue generation (Fan et al., 2018; Liu et al., 2020; 2022; Mirshekari et al., 2021; Gupta et al., 2021). Both of these two facts require the length of generation in LLMs can be effectively controlled.

For LLMs, the most widely applied technique for length control is prompt-based fine-tuning (Raffel et al., 2020; Goyal et al., 2022; Zhang et al., 2022; Liu et al., 2023a). Taking an example of length-controlled summarization (LCS), we can prepend a prompt "`summarize with length $l_i$:`" to the article to be summarized in training, where $l_i$ is the number of words or tokens of the reference summary. However, this process is usually performed in supervised fine-tuning (SFT), where this length controllable ability has to compromise with the goodness of downstream tasks. For very large LMs like GPT-3, the length controlled generation can be somewhat activated by in-context learning without updating the model parameters (Brown et al., 2020; Chowdhery et al., 2022; Dong et al., 2022), but this relies on the size and power of the pre-trained fundation models to achieve high control accuracy. For methods like RLHF (Reinforcement Learning from Human Feedback) (Christiano et al., 2017; Stiennon et al., 2020; Ouyang et al., 2022), it is expensive to use human for labelling whether the length of generated texts meets the requirement given in instructing prompts.

In general, there are many other length control methods such as GOLC, LenAtten and LAAM (Liu et al., 2018; Takase and Okazaki, 2019; Makino et al., 2019; Yu et al., 2021; Liu et al., 2022). However, these methods are not designed for pretrained LLMs, thus pre-training or different architectural designs are usually needed. Moreover, it is hard for existing length control methods to adapt to various precise control instructions such as greater than a target value, smaller than a target value, or

between two target values, etc. Therefore, how to effectively connect diverse control instructions from users to the final length of generated text for pretrained LLMs is still an issue to be tackled.

In this study, we introduce a novel method that applies prompt-based fine-tuning with reinforcement learning to improve the accuracy of length controlled generation. The main contributions are:

- We design a rule-based reward model for multiple control types other than traditional "equal to" control type, which can provide accurate and fast implementation for both reinforcement fine-tuning and inference of LLMs.

- We introduce an independent standard prompt extractors (SPE) to parse the length control instructions from diverse user inputs to standard control prompts (SCP), which is necessary for rule-based reward and show strong generalization power for new control prompts.

- We apply a Proximal Policy Optimization (PPO) algorithm with a modified state space to fine-tune LLMs for enhancing its length control ability. Two modes including (a) SCP + rule-based reward; (b) SCP + model-based reward are introduced and compared.

- Experiments show that by joint application of reinforcement fine-tuning and sample filtering, the length-control errors can be significantly reduced from the baseline prompt-based method. Moreover, the method show strong generalization ability to new prompt templates.

## 2 RELATED WORK

### 2.1 REINFORCEMENT LEARNING FOR TEXT GENERATION.

Reinforcement learning (RL) (Kaelbling et al., 1996; Arulkumaran et al., 2017) has been widely studied and applied to improve generation task performance, including summarization (Stiennon et al., 2020; Paulus et al., 2018), question generation (Pang and He, 2021), machine translation (Wu et al., 2016; Nguyen et al., 2017; Kiegeland and Kreutzer, 2021) and dialogue generation (Li et al., 2016; Zhou et al., 2017; Jaques et al., 2020). In general, we can consider the generative model as the policy network and optimize its parameters for achieving higher reward from the environment (Paulus et al., 2018; Wang et al., 2022). Human feedback is one of the most known strategies to get the reward, which is shown to be more effective than optimizing using some automatic metrics, such as rouge scores in text generation (Christiano et al., 2017; Stiennon et al., 2020; Wu et al., 2021). Existing study (Ramamurthy et al., 2023) also shows that RL techniques are generally better than supervised methods at aligning language models to human preferences. It is recently known that Reinforcement learning from Human Feedback (RLHF) plays a vital role in the success of autoregressive LLMs like InstructGPT (Ouyang et al., 2022), which utilizes human feedbacks on model generation and to train a reward model, and use it to align the LLMs with human intention through PPO reinforcement learning technique (Schulman et al., 2017).

### 2.2 LENGTH CONTROL FOR TEXT GENERATION

Length control is an important ability for text generation, especially for tasks with a large variance of output length, such as writing an article within a given length or summarizing texts using a desired range of number of words/tokens. Early work (Fan et al., 2018) on controlling lengths in abstractive summarization quantizes summary length into discrete bins, and expands the input vocabulary with special tokens to indicate the length bins of the ground-truth summary during training. Liu et al. (2018) extends a convolutional sequence to sequence model to control the length of summarization. To generate summaries of any desired length, a length constrain factor is added to each convolutional block of the initial layer. Takase and Okazaki (2019) proposes an extension of a sinusoidal positional encoding to enable neural encoder-decoder model to generate a text of any desired length. GOLC (Makino et al., 2019) dedicates to increase the probabilities of generating a high quality summary within a desired length by using minimum risk training. LenAtten (Yu et al., 2021) introduces a length attention unit to break the trade-off between length controllability and summary quality. LAAM (Liu et al., 2022) modifies the attention matrix based on length-budget dynamically during the decoding process. Generally, we notice that existing length control approaches can not be directly applied for control targets other than "equal to" a certain length, and are in lack of focusing on prompt-based method for the most recent trend of GPT-style LLMs.

# 3 METHOD

This study aims to investigate the length-controlled generation in LLMs, particularly for summarization, for which we propose a prompt-based method with the use of reinforcement learning and sample filtering. We first introduce the whole architecture and then discuss each component of it.

## 3.1 MODEL ARCHITECTURE

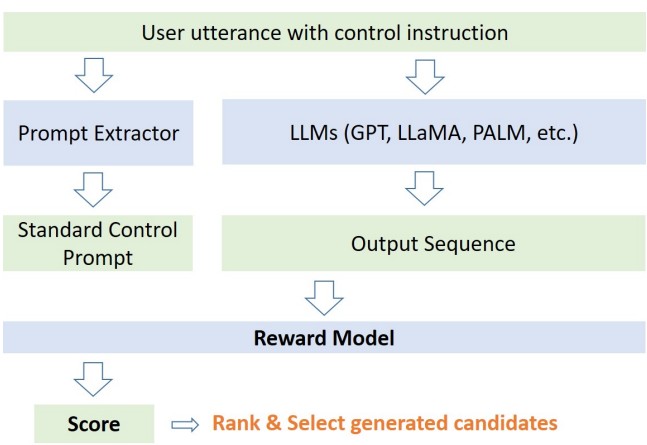

Figure 1: Overview of the model architecture. In training stage, the scores given by the reward model are used for the reinforcement learning method. In inference stage, the scores are applied for ranking and selecting the output sequences generated by LLMs.

The architecture of our model is given in Figure 1. The original user utterances may involve the control instruction on length constraint, which differs from factual and semantic information in terms of that it can be evaluated with rule-based methods. For instance, if we can somehow understand user intention on length constraint, we can set up this rule for ranking and selecting generated candidates. Therefore, we introduce a standard prompt extractor (SPE) to parse the information of length constraint from user utterance and thus generate a standard length control prompt. This standard prompt includes different types of length constraint and can be further applied for rule-based inference and evaluation (See Section 3.3).

As shown in Figure 1, the user utterance is passed through both a SPE and LLMs like GPT-family (Brown et al., 2020; OpenAI, 2023), PALM (Chowdhery et al., 2022; Anil et al., 2023), LLaMA (Touvron et al., 2023), Pangu (Ren et al., 2023), Ernie (Sun et al., 2019; 2020), etc. LLMs are the core modules that generate an output sequence according to the user utterance. The reward model takes both the standard control prompt (SCP) and generated sequence as input, and outputs a score to evaluate how well the generated sequence meets the requirement of this control prompt (See Section 3.2). The score can be applied as the reward signal in reinforcement learning method to fine-tune LLMs (See Section 3.4), or be applied to rank and select the generated sequences in inference (see Section 3.5).

## 3.2 REWARD MODEL

| Standard Control Prompt | Reward |
|---|---|
| more than $L_t$ | $-\text{ReLU}(L_t - L_g)$ |
| less than $L_t$ | $-\text{ReLU}(-L_t + L_g)$ |
| equal to $L_t$ | $-\lvert L_t - L_g \rvert$ |
| between $L_L$ and $L_U$ | $-(\text{ReLU}(L_L - L_g) + \text{ReLU}(L_g - L_U))$ |

Table 1: Standard control prompts (SCPs) with corresponding reward functions.

To evaluate whether the generated text meets the requirement of length control instruction, we introduce a reward model to score the generated sequences based on the length constraint in user utterances. This score can be used as a reward for fine-tuning existing LLMs by leveraging reinforcement learning, or be used to rank and select the candidates generated by LLMs. We propose **rule-based reward models**, in which we use a SPE to parse each user utterance and get its type of length constraint and target values as described in Table 1 and Section 3.3. Using the actual length of the output sequence, we can finally calculate the rewards based on the right column of Table 1, where $L_t$, $L_L$, $L_U$ and $L_g$ refer to the target length, the lower-bound length, the upper-bound length and the actual generated length, respectively. The advantage of rule-based method is that it provides the accurate evaluation on lengths given the SCP, while the latency is almost negli-

gible compared with using BERT or GPT models for scoring. However, it relies on extracting exact standard control information from the user's input. We also discuss the use of model-based reward models in Appendix A.5.3.

## 3.3 STANDARD PROMPT EXTRACTOR

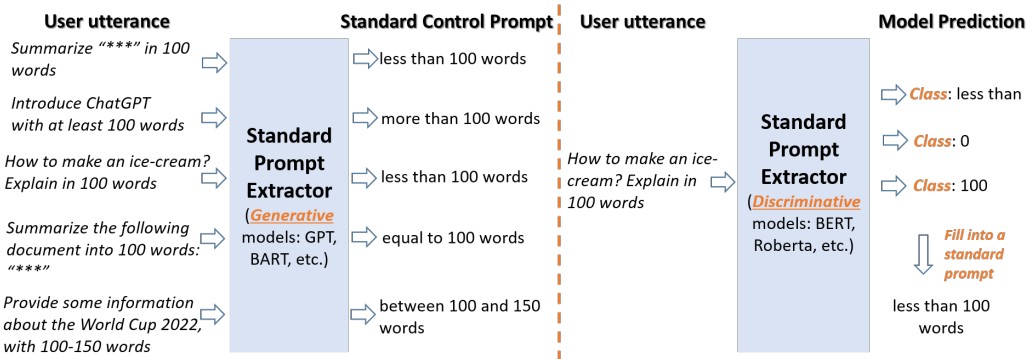

Figure 2: The demonstration of Standard Prompt Extractor (SPE). The generative type of models are trained to output the standard control prompts (SCPs) directly (left), while the discriminative type of models are trained to predict the type of each control instruction, as well as the requested number of lengths from user utterance, such as the minimum value and the maximum value (right).

As above discussed, to get SCPs for applying rule-based reward model to score the generated sequences in RL and sample filtering, we introduce standard prompt extractor (SPE). It takes a user utterance as input, and outputs the SCP if exists. This standard prompt consists of a basic description of what length constraint should be satisfied. As is shown in Figure 2, the prompt extractor can be a **generative model** such as GPT, in which case the extractor is trained directly to generate the full SCP as is shown by Figure 2 (left). The final control signal can be parsed into $L_t$, $L_U$ and $L_L$ as in Table 1. We can also use a **discriminative model** such as BERT, as the prompt extractor, in which case it is required to predict the type of SCP and the target numbers involved, as is shown in Figure 2 (right). In this case, we prepend three [CLS] tokens at the beginning of the input. Three linear projection layers with different output sizes (i.e., number of types of control instruction, number of possible minimum values, number of possible maximum value) map the three top vectors of [CLS] tokens to fill in the type, minimum value and maximum value of a standard prompt template. Therefore, we have three classification targets based on the three top vectors for predicting the ground truth of SCP. Also, we can just use the minimum and maximum target values without type information, where two [CLS] tokens and corresponding linear projections are needed.

## 3.4 REINFORCEMENT LEARNING FOR LENGTH CONTROL FINE-TUNING

We apply a modified PPO method with actor-critic setting (Grondman et al., 2012; Bahdanau et al., 2017; Schulman et al., 2017). Since evaluating the generated length does not depend on the input article, both the reward model and critic model only take the concatenation of the SCP and the generated text as input. As the reward for length control can only be determined after the end of generation, we only calculate the reward and advantage with the final output. Assume $\pi_\theta(a|s)$ is a stochastic policy given by the GPT model, where $\theta$ is the trainable parameter, $s$ is the whole input sequence, and $a$ is the finally generated sequence. The original policy gradient (PG) applied the loss function given by Equation 1.

$$L^{PG}(\theta) = -\hat{\mathbb{E}}_D[\log \pi_\theta(a|s)\hat{A}], \tag{1}$$

where $\hat{\mathbb{E}}_D[.]$ is the empirical average over a finite batch of samples from dataset $D$. $\hat{A}$ is an estimator of the advantage function at the end of generation. For the actor-critic case, we set $\hat{A} = R(s', a) - \hat{V}_{\phi_{old}}(s', a)$, where $R(.)$ is the reward model, $\hat{V}_{\phi_{old}}(s', a)$ is the expect value from the critic model of the last step. Note that the value and reward only depend on the standard control prompt $s'$ and the generated sequence $a$. However, the original PG empirically often leads to a large policy update

and thus instability during fine-tuning. Therefore, both the trust region method (TRPO) (Schulman et al., 2015) and Proximal Policy Optimization (PPO) (Schulman et al., 2017) use the probability ratio $r(\theta) = \frac{\pi_\theta(a|s)}{\pi_{\theta_{old}}(a|s)}$ instead of $\log \pi_\theta(a|s)$ in Equation 1. PPO utilizes a clipped surrogate objective given by Equation 2 to stabilize the policy updates and ensure that the probability ratio term is bounded between $[1 - \epsilon, 1 + \epsilon]$.

$$L^{CLIP}(\theta) = -\hat{\mathbb{E}}_D[\min(r(\theta)\hat{A}, \text{Clip}(r(\theta), 1 - \epsilon, 1 + \epsilon)\hat{A})]. \tag{2}$$

To ensure sufficient exploration, we follow the PPO method (Schulman et al., 2017) to introduce an entropy term $S = \frac{1}{n}\sum \pi_\theta(a|s)\log(\pi_\theta(a|s))$, in which the average is taken across the vocabulary dimension. In addition, we add a penalty for large KL divergence between the current and old stochastic policy distributions (i.e. $\pi_\theta$ and $\pi_{\theta_{old}}$). Therefore, the total policy loss to be optimized can then be rewritten as:

$$L^{CLIP+S+KL}(\theta) = \hat{\mathbb{E}}_D[L^{CLIP}(\theta) - cS[\pi_\theta|(s)] + \beta D_{KL}(\pi_\theta|\pi_{\theta_{old}})], \tag{3}$$

where $c, \beta$ are coefficients, $D_{KL}(\pi_\theta|\pi_{\theta_{old}})$ is the KL-divergence between the old and current action probability distributions. To avoid the performance loss for downstream task, we add an extra terms of SFT loss from the same batch of labeled data on the actor's policy loss: $L^A(\theta) = L^{CLIP+S+KL}(\theta) + \lambda L^{SFT}(\theta)$, where $\lambda$ is a tunable hyper-parameter. Meanwhile, we optimize a value loss $L^{VF} = (V_\phi(s', a) - \hat{R})^2$. The detailed algorithm is given in Appendix A.

## 3.5 INFERENCE & SAMPLE FILTERING

In the inference stage, the well fine-tuned LLMs can directly process user utterances and generate a sequence following the expected length control instructions of user intention. This relies on the generalization ability of the model if the control information in the user input is in diverse expressions. In another word, our proposed prompt extractor serves as an important role to parse the SCP to benefit the latter RL fine-tuning. Based on this, we can apply either trainable or rule-based reward model to score, rank and select from a set of generated samples in beam sampling, which is named as sample filtering in our method. Let $k = \arg\max_i R(s', a_i)$, where $R$ is the reward model, $a_i$ is the $i$th sequence in all $N$ output sequences, then a $a = a_k$ is selected to be the final output sequence. Thereafter, the selected sequence can be used for both the RL fine-tuning phase and the final evaluation to judge to what extent the length control ability can be achieved in existing LLMs.

## 4 EXPERIMENTS

### 4.1 EXPERIMENTAL SETUP

We perform experiments on two popular summarization datasets including CNNDM (Hermann et al., 2015) and NYT (Durrett et al., 2016). CNNDM contains news articles from the CNN and Daily Mail websites, with labelled abstractive and extractive summaries. There are 287,226 training samples, 13,368 validation samples and 11,490 test samples. NYT contains 110,540 articles with abstractive summaries from New York Times. We follow its paper to split the original dataset into 100,834 training and 9,706 test examples. This following subsections explain how to train and use different modules of our method. We leave the detailed setting of hyper-parameters in Appendix A.4.

### 4.1.1 DATA PROCESSING AND AUGMENTATION

We design a set of SCPs, including five types of control instructions: "more than **
tokens", "less than ** tokens", "equal to ** tokens", "between ** and
** tokens" and "none". "**" means the expected length value of user intention, and "none"
means no length constraints. For each type, we randomly sample a target summary length from 50 tokens to 150 tokens based on the general news summary length, and fill these lengths into "**"
field of a randomly sampled SCP. For further simulate real user utterances with length control intention, about 10-20 different augmented prompt templates are applied for each SCP. The examples of templates are shown in Figure 2 and Appendix A.3. Finally, we can create augmented input data by replacing the placeholders in the augmented templates with target numbers and original articles.

### 4.1.2 TRAINING OF STANDARD PROMPT EXTRACTOR

As discussed in Section 3.1, we train two types of models, *i.e.,* generative and discriminative models, to serve as a standard prompt extractor. In particular, we fine-tune a GPT-style model (GPT2-small) as a generative extractor and a BERT-small model as a discriminative extractor. Both pre-trained checkpoints are obtained from huggingface (Wolf et al., 2019). We use the augmented input data as discussed in Section 4.1.1. To make it clear, we use the original articles of CNNDM and NYT, and first sample a SCP for each article, and then sample an augmented prompt template from a pre-designed set. Next, we randomly assign the target length values between 50 and 150 to each article to form the finalized augmented template. Each original article associated with its augmented template serves as input data, and its corresponding SCP serves as the expected prediction, to finally train the standard prompt extractor.

| Extractor | Acc. | Acc. Gen. |
|---|---|---|
| BERT-base-cls-2 | **100.0** | **100.0** |
| BERT-base-cls-3 | 99.7 | 99.8 |
| GPT-small | 97.7 | 97.5 |

Table 2: Evaluation on the accuracy and generalization of standard prompt extractors (SPEs). "cls-2" and "cls-3" refer to only predicting the minimum and maximum values, or predicting the control type as well. The "Acc. Gen." column denotes the generalization performance of SPE on unseen prompt templates in test set.

Experimental results on evaluating SPEs are given in Table 1. "Acc." is the prediction accuracy on test set, and "Acc. Gen." means we apply 30% of randomly sampled augmented control prompts as out-of-sample templates for evaluation, and only train the SPE model on the remaining 70% templates. Results show that BERT-base-cls-2 models can be trained to achieve 100.0% test accuracy for extracting SCPs, and it also generalizes well for out-of-sample control prompts that are not used in training. The accuracy of GPT-small is relatively lower, which may because fully matching the whole generated text strings is more difficult than extracting the key values. Details of the learning curves are provided in Appendix A.6. In general, we believe well selected extra extraction module does not introduce much noise or accuracy loss in end-to-end implementation with rule-based reward models. We use BERT-base-cls-2 discriminative extractor in later experiments to get clear and accurate minimum and maximum target values.

### 4.1.3 SUPERVISED FINETUNING OF GPT MODELS

To build the baseline summarization model with length control ability, we apply three pre-trained GPTs with 124M, 355M and 774M parameters from Huggingface, denoted as GPT-S, GPT-M, GPT-L, respectively. We experiment on two types of control settings. The first one is **single-type control**, where we only consider the strict SCP of "`equal to`". In details, for each example we randomly sample a augmented control prompt under the type of "equal" and replace the text placeholder with the input text and replace the length placeholder with the real text length of reference summary. The second setting is **multiple-type control**, in which we randomly split the original dataset into four parts, and each is augmented with one type of SCP. We then compare the real text length of each reference summary with one (for "`less than **`" or "`more than **`") or two (for "`between ** and **`") randomly sampled target lengths between 50 and 150. Similar to the single-type control, we replace "`**`" with the corresponding sampled length. Finally, we perform SFT on the labelled data to enable pre-trained GPTs to summarize texts with a length control ability.

### 4.1.4 FINETUNING WITH REINFORCEMENT LEARNING

Based on the supervised fine-tuned LLMs in the above, we propose to further improve the accuracy of length control via reinforcement learning with the PPO method described in Section 3.4. We consider two settings to generate the reward. The first is to process the augmented inputs with SCP and use a rule-based reward model based on the extracted standard control information, the second is to apply a trainable reward model. Exploratory experiments show that actor-critic generally works better than actor-only as shown in Table 14 in Appendix, thus in the main experiments we use actor-critic setting. We apply AdamW optimizers with $\beta_1 = 0.9$, $\beta_2 = 0.999$ for both the actor and critic (if applied) model, while the learning rate is set to 3e-7 for actor model and 3e-4 for critic model (if applied). No learning rate schedule is used, and weight decay is set to 0. For each iteration, the policy is run for every 512 samples to estimate the reward and value. In surrogate optimization of

each iteration, we set the epoch number to 16 and the mini-batch size to 8. The clipping parameter $\epsilon$ in Equation 2 is 0.2, weights parameters in Equation 3 is set to $c = 0.01$ and $\beta = 0.1$.

## 4.2 RESULTS

| Model | Setting | CNNDM | | | | | NYT | | | | |
|-------|---------|------|------|------|------|------|------|------|------|------|------|
| | | R1↑ | R2↑ | RL↑ | B.S.↑ | Error↓ | R1↑ | R2↑ | RL↑ | B.S.↑ | Error↓ |
| GPT-S | Prompt | 37.57 | 15.30 | 37.74 | 62.47 | 11.62 | 47.48 | 29.27 | 42.36 | 67.86 | 13.33 |
| | Prompt+RL | 37.44 | 15.02 | 39.05 | 62.10 | 7.81 | 47.59 | 29.41 | 42.66 | 67.82 | 11.92 |
| | Prompt+filter | 38.20 | 16.02 | 37.31 | 61.96 | 10.44 | 48.37 | 30.83 | 42.72 | 67.96 | 10.30 |
| | Prompt+RL+filter | 37.56 | 15.85 | 38.47 | 61.53 | **6.22** | 48.31 | 30.94 | 42.82 | 67.98 | **9.55** |
| GPT-M | Prompt | 38.05 | 16.15 | 37.81 | 62.93 | 14.31 | 48.34 | 30.53 | 43.11 | 68.54 | 5.12 |
| | Prompt+RL | 37.73 | 15.98 | 38.07 | 62.62 | 11.57 | 48.86 | 31.19 | 43.98 | 69.09 | 4.47 |
| | Prompt+filter | 38.18 | 16.55 | 37.14 | 62.32 | 12.60 | 48.53 | 30.95 | 43.33 | 68.55 | 2.12 |
| | Prompt+RL+filter | 37.91 | 16.33 | 36.97 | 62.23 | **11.33** | 48.76 | 31.09 | 43.38 | 68.80 | **1.60** |
| GPT-L | Prompt | 40.27 | 17.33 | 39.67 | 63.96 | 12.20 | 49.98 | 32.43 | 44.65 | 69.44 | 5.89 |
| | Prompt+RL | 39.49 | 16.42 | 39.02 | 63.38 | 9.84 | 49.12 | 30.86 | 43.59 | 69.03 | 5.54 |
| | Prompt+filter | 39.52 | 17.33 | 38.64 | 63.22 | 11.57 | 47.22 | 31.77 | 43.29 | 69.02 | 5.76 |
| | Prompt+RL+filter | 39.75 | 17.18 | 38.60 | 63.15 | **8.96** | 49.82 | 31.68 | 42.48 | 68.72 | **3.29** |

Table 3: Comparison of methods in the setting of single-type control instruction, i.e., "`equal to`".

| Model | Setting | CNNDM | | | | | NYT | | | | |
|-------|---------|------|------|------|------|------|------|------|------|------|------|
| | | R1↑ | R2↑ | RL↑ | B.S.↑ | Error↓ | R1↑ | R2↑ | RL↑ | B.S.↑ | Error↓ |
| GPT-S | Prompt | 37.76 | 15.58 | 38.05 | 62.32 | 18.16 | 47.22 | 29.47 | 42.01 | 67.76 | 31.15 |
| | Prompt+RL | 37.52 | 15.31 | 38.79 | 62.42 | 14.29 | 47.30 | 29.84 | 42.36 | 67.81 | 10.53 |
| | Prompt+filter | 38.04 | 16.29 | 37.12 | 62.05 | 10.57 | 47.88 | 30.55 | 42.50 | 67.87 | 8.06 |
| | Prompt+RL+filter | 37.48 | 16.01 | 37.20 | 61.88 | **7.06** | 47.84 | 30.43 | 42.26 | 67.54 | **3.89** |
| GPT-M | Prompt | 38.85 | 15.93 | 38.48 | 63.02 | 21.32 | 48.34 | 30.74 | 43.64 | 68.75 | 13.17 |
| | Prompt+RL | 38.30 | 15.89 | 39.29 | 62.90 | 6.59 | 48.23 | 30.58 | 43.61 | 68.67 | 12.61 |
| | Prompt+filter | 38.85 | 17.29 | 37.68 | 62.48 | 11.21 | 49.73 | 32.65 | 44.55 | 69.00 | 6.75 |
| | Prompt+RL+filter | 37.83 | 16.89 | 37.20 | 61.91 | **4.98** | 49.41 | 32.18 | 44.05 | 68.40 | **3.65** |
| GPT-L | Prompt | 38.27 | 16.37 | 38.92 | 63.09 | 6.89 | 49.41 | 32.20 | 44.31 | 69.36 | 10.64 |
| | Prompt+RL | 38.23 | 16.42 | 38.86 | 63.06 | 6.62 | 49.35 | 32.24 | 44.31 | 69.27 | 8.52 |
| | Prompt+filter | 38.75 | 16.85 | 38.23 | 62.85 | 3.34 | 50.04 | 32.65 | 44.35 | 69.48 | 4.82 |
| | Prompt+RL+filter | 38.70 | 16.52 | 38.39 | 62.98 | **3.22** | 50.01 | 32.52 | 44.14 | 69.51 | **4.60** |

Table 4: Comparison of methods in the setting of multiple-type control, in which we consider all the four candidate types of control instructions, as shown in Table 1.

### 4.2.1 MAIN RESULTS

As Table 3 shows, we compare models with four different settings for prompt-based length control, including (1) **Prompt**: use GPTs after prompt-based SFT in Section 4.1.3 to control the output length; (2) **Prompt+RL**: the GPTs used in (1) but further enhanced with reinforcement learning; (3) **Prompt+filter**: the LLM in (1) but equipped with sample filtering; and (4) **Prompt+RL+filter**: the enhanced GPTs with both RL and sample filtering, which is a combination of (2) and (3). Other existing methods such as LenAtten and LAAM (Yu et al., 2021; Liu et al., 2022) apply different length distributions and base models, and are not adaptive to prompt based length control with multi-type control instructions for GPTs. Thus, we do not report the results in their papers for comparison. For evaluation, we apply relevance scores including F1 of Rouge Scores (ROUGE, 2004) (denoted as "R1", "R2", "RL") and BertScore (Zhang et al., 2019) (denoted as "B.S"), and control error (denoted as "Error") which is the negative reward in Table 1 representing the average difference between the output length and the desired range. We select the checkpoint with the lowest validation control error and less than 1 point's drop of BERTScore for evaluation on the test set. For all methods with sample filtering, we set the number of output sequences to 8, and select the one with the highest reward. Results of the single-type control that only considers "`equal to`" are given in Table 3, while the results of multi-type control using augmented input with all the SCPs (see Table 1) are presented in Table 4. Note that Rouge scores and BERTScore can be less

than the general state-of-the-art summarization models without random sampled target length, since the length sampling distributions can be very different from the reference summaries. In fact, the mean and standard deviation of the labelled summary lengths are 71 and 28 tokens respectively for CNNDM, 104 and 35 tokens for NYT. The difference of control errors for two datasets may partly be due to this length distribution. Overall, we can see that for all settings, the proposed RL method can provide an improvement of length control ability with lower control errors. By further using sample filtering supported by the rule-based reward model, both the basic prompt-based length control model `Prompt+filter` and the one with RL enhancement `Prompt+RL+filter` can achieve lower control errors than not using sample filtering like the method (1) and (2). After checking the learning curves (see Appendix A.7), we also find that the relevance metric BertScore indeed does not have a clear decrease trend in early stage as the validation reward increases.

## 4.3 COMPARING OF DIFFERENT CONTROL TYPES

| Control | Setting | R1 | R2 | RL | B.S. | Error↓ |
|---|---|---|---|---|---|---|
| Equal | Prompt | 38.1 | 15.7 | 38.9 | 62.6 | 26.1 |
| | +RL | 35.7 | 14.6 | 38.7 | 61.9 | 13.6 |
| | +filter | 37.9 | 16.3 | 37.4 | 61.9 | 12.5 |
| | +RL+filter | 37.6 | 16.1 | 38.2 | 62.2 | **8.4** |
| Less | Prompt | 37.1 | 14.7 | 36.6 | 61.9 | 0.5 |
| | +RL | 37.4 | 14.8 | 37.0 | 62.1 | 0.4 |
| | +filter | 36.9 | 15.7 | 35.9 | 61.1 | **0.2** |
| | +RL+filter | 36.9 | 15.7 | 35.9 | 61.1 | **0.2** |
| More | Prompt | 38.0 | 15.4 | 37.8 | 62.4 | 41.9 |
| | +RL | 35.8 | 14.8 | 38.9 | 61.8 | 13.8 |
| | +filter | 38.5 | 16.4 | 37.6 | 62.1 | 23.1 |
| | +RL+filter | 37.4 | 16.3 | 37.9 | 62.2 | **6.0** |
| Between | Prompt | 36.4 | 15.0 | 38.7 | 62.0 | 5.8 |
| | +RL | 36.1 | 15.0 | 39.0 | 61.8 | 4.5 |
| | +filter | 38.1 | 16.4 | 37.4 | 62.1 | 1.2 |
| | +RL+filter | 37.9 | 16.3 | 37.4 | 62.0 | **1.1** |

Table 5: Comparison of four control types in the multiple-type control setting using GPT-S on CNNDM.

We de-construct the setting of multiple-type control and thus evaluate the effect of our proposed method on each particular control type. Results on CNNDM are given in Table 5. Note that here we apply the SFT on GPT-small model for multiple-type control setting as like Table 4, so the errors of type "`equal to`" can be different from Table 3. Therefore, in this ablation study, the baseline *Prompt* gets a higher control error due to that it involves multiple control types and owns a more complex training goal. In general, our proposed methods bring a significant improvement of length control accuracies (i.e., Error) for all the four control types. Moreover, some insightful findings can be obtained from Table 5. As the average length of labelled summary in CNNDM (71 tokens) is much less than the average of sampled target lengths, i.e., 100 tokens, therefore, to generate with "`more than`" a sampled target length is harder than "`less than`" for all candidate methods. However, the `Prompt+RL+filter` can still provide a significantly large improvement on the control type of "`more than`", by reducing the Error from 41.9 to 6.0. In the case of "`less than`" with sample filtering, the RL method does not further reduce the validation error as it is already quite low, thus the default checkpoint is always selected even after RL fine-tuning. We also provide this ablation study on NYT in Table 13 in Appendix, and similar results and insights can be observed.

## 4.4 GENERALIZATION TO OUT-OF-SAMPLE PROMPT TEMPLATES

| Type | Setting | R1 | R2 | RL | B.S. | Error↓ |
|---|---|---|---|---|---|---|
| SG | Baseline | 37.6 | 15.3 | 37.7 | 62.5 | 11.6 |
| | In-sample | 37.4 | 15.0 | 39.1 | 62.1 | 7.8 |
| | Out-sample | 36.7 | 15.0 | 39.1 | 61.3 | 8.0 |
| MU | Baseline | 37.8 | 15.6 | 38.1 | 62.3 | 14.7 |
| | In-sample | 37.5 | 15.3 | 38.8 | 62.4 | 8.9 |
| | Out-sample | 37.9 | 15.7 | 38.9 | 62.5 | 9.6 |

Table 6: Generalization to out-of-sample control prompt templates of GPT-S on CNNDM.

To evaluate if the tuned model can generalize to unseen prompt templates of length control, we conduct an extra experiment by tuning on a 70% subset of prompt templates randomly sampled from Table 8, and check the generalization performance of the model on test data with the rest unseen prompt templates. The results are give in Table 6, where the difference between "In-sample" and "Out-sample" setting is if an out-of-sample set of control prompt templates in Table 8 are applied for augmenting the test dataset in CNNDM.

We notice that in some cases, there is a slight performance degradation on out-of-sample prompt templates, but the length control ability is still significantly better than baseline prompt-based method. This demonstrates that the propose method has strong generalization ability to novel prompt templates. We believe with a larger set of prompt templates in training, the generalization power can still be largely improved.

## 4.5 Comparing of different reward models

We further perform an empirical study to compare the performance of rule-based reward model and other trainable reward models including BERT-large and GPT-medium. For BERT, we map the `[CLS]` token vector to a single score value via a linear projection layer. For GPTs, we add a linear projection layer on top of the last token vector to predict the score. The training is to minimize the Mean Squared Error (MSE) loss between the predicted scores and the labelled scores for length control. We build a simulated dataset with 100,000 examples using the original CNNDM and NYT datasets by concatenating sampled standard control

| Setting | R1 | R2 | RL | B.S. | Error↓ |
|---|---|---|---|---|---|
| Prompt | 37.4 | 15.2 | 37.6 | 62.3 | 11.9 |
| +RL (Rule) | 37.3 | 14.9 | 38.9 | 61.8 | **7.4** |
| +RL (GPT) | 37.7 | 14.9 | 38.2 | 62.0 | 9.3 |
| +RL (BERT) | 37.4 | 15.0 | 38.3 | 70.2 | 9.1 |
| +filter | 38.3 | 16.1 | 37.4 | 61.9 | 10.5 |
| +RL+filter (Rule) | 37.3 | 15.7 | 38.7 | 61.2 | **6.3** |
| +RL+filter (GPT) | 37.1 | 15.7 | 37.9 | 61.5 | 8.8 |
| +RL+filter (BERT) | 36.6 | 15.1 | 37.0 | 69.2 | 7.8 |

Table 7: Results by using different types of reward models (Rule-based; GPT; BERT) for our method in the single-type control setting ("`equal to`"). GPT-S and CNNDM are used.

prompts with randomly sampled target lengths from 50 to 150 and the summaries from the labels. The score labels are then calculated by comparing the real lengths of labeled summaries and the sampled control prompts by using Table 1. Then we fine-tune the pre-trained GPT-medium or BERT-large (from Huggingface) as the trainable reward models to predict the reward labels by minimizing MSE losses. We consider the single-type control setting and apply the modified PPO algorithm with the above reward models to fine-tune GPT-small models to perform length-control summarization on the augmented data based on CNNDM. Experimental results on the selected checkpoint with the best control accuracy for each setting are shown in Table 7. We observe that all of three settings using reinforcement learning achieve a significantly lower control error than the baseline model only using the prompt-based strategy (i.e., `Prompt`) on CNNDM. In addition, the setting of reinforcement learning with rule-based reward model, i.e., `Prompt+RL(Rule)`, generally outperforms other models in terms of control error. It validates the effectiveness of rule-based reward models in the RL fine-tuning. Extra results on NYT show that the GPT-based reward model can slightly outperform the BERT-based one. The details are provided in Table 16 in Appendix.

## 5 Conclusion

The paper proposes a method for improving the length control ability of GPT-style LLMs, especially for the domain of text summarization, due to that this task usually has a larger variance of output lengths and more standard automatic metrics than other alternatives. The standard prompt extractor and the rule-based reward model are introduced to provide an accurate control signal for both fine-tuning and inference. A modified PPO algorithm with a state space particularly designed for length control is applied for enhancing the length controlled generation. The method is shown to be effective for three GPTs with different sizes on both CNNDM and NYT summarization datasets. Compared to the baseline method using prompt-based strategies on GPTs, our method achieves a significant improvement in terms of control accuracy. Moreover, it can process diverse length-control prompts with strong generalization power to new prompt templates, which is not well handled by existing length control methods. Our method can be applied in a variety of LLMs to improve the user experience by outputting the text with a desired length. We believe for other format control (e.g., rhyme scheme in a poem or song) that can be somehow evaluated or scored by a rule-based reward model, our proposed method can also be applied. Meanwhile, the limitation of our method is that it does changes the parameters of the pretrained models, which may result in a risk of performance loss in some cases. Well designed in-context learning or introducing adaptors particularly tuned for length control may be potential solutions for this.

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

## A   APPENDIX

### A.1   ALGORITHM FOR LENGTH CONTROLLED FINE-TUNING WITH OUR MODIFIED PPO

Following the explanations in Section 3.4, we further provide an algorithm table for our modified PPO fine-tuning in Algorithm 1.

---

**Algorithm 1:** Algorithm for controlled fine-tuning with modified PPO

---

1: Get a pretrained GPT model to initialize the policy network $\pi_{\theta_{old}}(a|s)$.
2: Initialize critic network $V_\phi(s', a)$.
3: Initialize hyper-paramaters $N_{iteration}, M, B, n_{epoch}, c, \beta$.
4: **for** i<=1,...,$N_{iteration}$ **do**
5:    **for** j=1,...,M **do**
6:       Get an input sequence $s_0$ augmented with random sampled augmented control prompt from the data-loader.
7:       Run SPE to get the SCP $s'$ from the input sequence.
8:       Run policy $\pi_{\theta_{old}}(a|s)$ for an input sequence with augmented control prompt $s$ to get an output sequence $a$, policy $\pi_{\theta_{old}}$.
9:       Get the reward of output sequence $a$ with reward model $r = r(s', a)$.
10:      Store input $s$, SCP $s'$, generate sequence $a$, reward $r$ and old policy $\pi_{\theta_{old}}$ into memory.
11:    **end for**
12:    **for** e=1,...,$n_{epoch}$ **do**
13:       **for** b=1,...,B **do**
14:         Take the $b$-th mini-batch $(s', a, r, \pi_{\theta_{old}})$ from the memory.
15:         Use the actor and critic networks to get the new policy and value $\pi_\theta(a|s), V_\phi(s', a)$.
16:         Compute the ratio $r(\theta) = \frac{\pi_\theta(a|s)}{\pi_{\theta_{old}}(a|s)}$.
17:         Compute advantage estimate $\hat{A} = r - V_{\phi_{old}}(s', a)$.
18:         Compute $L^{CLIP}$ with Equation 2.
19:         Compute the KL-divergence $D_{KL}(\pi_\theta|\pi_{\theta_{old}})$.
20:         Compute the Entropy $S[\pi_\theta|(s)]$.
21:         Compute the actor loss $L_\theta^A$ with Equation 3.
22:         Update the policy network parameters $\theta$ with gradients of $L_\theta^A$.
23:         Compute the value loss $L_\phi^V = MSE(V_\phi(s', a), r)$.
24:         Update the critic network parameters $\phi$ with gradients of $L_\phi^V$.
25:       **end for**
26:    **end for**
27: **end for**
28: **return** $\theta$

---

### A.2   DETAILS FOR DATASETS

CNNDM contains news articles from the CNN and Daily Mail websites, with labelled abstractive and extractive summaries. There are 287,226 training samples, 13,368 validation samples and 11,490 test samples. NYT contains 110,540 articles with abstractive summaries from New York Times. We follow its paper to split the original dataset into 100,834 training and 9,706 test examples. When GPT-2 tokenizer is applied, the labeled summaries in CNNDM have an average length of 71 tokens with a standard deviation of 28 tokens, while the labeled summaries in NYT have an average length of 104 tokens with a standard deviation of 28 tokens.

### A.3   EXAMPLES OF STANDARD CONTROL PROMPT AND AUGMENTED CONTROL PROMPT TEMPLATES

The SCPs and corresponding augmented prompt templates for generating the augmented input with length control information are given in Table 8. In the experiments, we use the augmented prompts

to train and evaluate the standard prompt extractor. For reward models and generation models, SCPs can be considered as available given a high-performing standard prompt extractor.

| Equal | Less | More | Between |
|---|---|---|---|
| summarize "*" with length ? | summarize "*" with length smaller than ? | summarize "*" with length larger than ? | summarize "*" with length between ! and ? |
| summarize the following document with length ?: "*" ' | summarize the following document with length smaller than ?: "*" | summarize the following document with length larger than ?: "*" | summarize the following document with length between ! and ?: "*" |
| Summarize with exactly ? tokens: *' | Summarize with less than ? tokens: * | Summarize with more than ? tokens: * | Summarize with between ! and ? tokens: * |
| I want a summary of "*" with exactly ? Tokens | I want a summary of "*" with less than ? Tokens | I want a summary of "*" with more than ? Tokens | I want a summary of "*" with between ! and ? Tokens |
| Give me a summary with ? tokens from "*"' | Give me a summary with less than ? tokens from "*" | Give me a summary with more than ? tokens from "*" | Give me a summary with between ! and ? tokens from "*" |
| Please summarize "*" with exactly ? Tokens | Please summarize "*" with less than ? Tokens | Please summarize "*" with more than ? Tokens | Please summarize "*" with between ! and ? Tokens |
| Write a summary of "*" with exactly ? Tokens | Write a summary of "*" with less than ? Tokens | Write a summary of "*" with more than ? Tokens | Write a summary of "*" with between ! and ? Tokens |
| summarize "*" with ? tokens for me | summarize "*" with less than ? tokens for me | summarize "*" with more than ? tokens for me | summarize "*" with between ! and ? tokens for me |
| Please give me a summary of "*" with ? Tokens | Please give me a summary of "*" with less than ? Tokens | Please give me a summary of "*" with more than ? Tokens | Please give me a summary of "*" with between ! and ? Tokens |
| I need a summary of length ? for "*" | I need a summary of length ? for "*" | I need a summary of length ? for "*" | I need a summary of length between ! and ? for "*" |
| generate a summary for "*" with length ? | I need a summary of length less than ? for "*" | I need a summary of length larger than ? for "*" | Need a summary of "*" with length between ! and ? |
| Need a summary of "*" with length equal to ? | Need a summary of "*" with length smaller than ? | Need a summary of "*" with length larger than ? | write a summary of length between ! and ? for "*" |
| write a summary of length ? for "*" | summarize the following article with no longer than ? tokens: "*" | summarize the following article with longer than ? tokens: "*" | summarize with length between ! and ?: "*" |
| summarize with length equal to ?: "*"' | summarize the following article with shorter than ? tokens: "*" | write a summary of length larger than ? for "*" | summarize with between ! and ? tokens:"*" |
| summarize with exactly ? tokens:"*" | write a summary of length smaller than ? for "*" | summarize with length larger than ?: "*" | summarize with ! to ? tokens:"*" |
| summarize this document with about ? tokens: "*" | summarize with length smaller than ?: "*" | summarize with more than ? tokens:"*" | summarize "*" with ! to ? Tokens |
| summarize "*" with around ? tokens | summarize with less than ? tokens:"*" | summarize the following article with over ? tokens:"*" | Please summarize "*" with ! to ? Tokens |
| need a summary of "*" with length ? | summarize "*" within ? tokens | summarize "*" with over ? tokens | summarize following article with ! to ? tokens: "*" |

Table 8: Examples of standard control prompts and corresponding augmented prompt templates, where each column shows one SCP followed by augmented prompt templates. Where "*" is the placeholder for input document to be summarized, "!" and "?" are the placeholders for the sampled length value. To build the input examples in training and evaluation datasets, we only need to first replace "!" and "?" with the minimum and maximum target lengths, and then replace "*" with the original article to be summarized.

## A.4 HYPER-PARAMETER SETTINGS

In this section, we provide hyper-parameter settings of different modules and training stages of our method, where we denote hyper-parameter as "HP" in the tables. For the standard prompt extractor, the hyper-parameter settings are given in Table 9. For the trainable reward models, the hyper-parameter settings are given in Table 10. For pretraining of GPT summarization models with control prompts, the hyper-parameter settings are given in Table 11. For enhancing control ability with reinforcement finetuning, the hyper-parameter setting are given in Table 12.

| HP | BERT extractor | GPT extractor |
|---|---|---|
| pretrained model | BERT-small | GPT-small |
| optimizer | AdamW | AdamW |
| batch size | 32 | 64 |
| lr | 2E-05 | 2E-05 |
| $\beta_1$ | 0.9 | 0.9 |
| $\beta_2$ | 0.999 | 0.999 |
| weight decay | 1E-07 | 0 |
| num iterations | 200k | 200k |

Table 9: Hyper-parameter setting of Standard Prompt Extractors.

| HP | BERT reward | GPT reward |
|---|---|---|
| pretrained model | BERT-large | GPT-medium |
| optimizer | AdamW | AdamW |
| batch size | 64 | 32 |
| lr | 0.00005 | 0.00005 |
| $\beta_1$ | 0.9 | 0.9 |
| $\beta_2$ | 0.999 | 0.999 |
| weight decay | 0 | 0 |
| num iterations | 200k | 200k |

Table 10: Hyper-parameter setting of trainable reward models.

| HP | GPT-S | GPT-M | GPT-L |
|---|---|---|---|
| optimizer | AdamW | AdamW | AdamW |
| batch size | 64 | 64 | 64 |
| lr | 5E-05 | 5E-05 | 2E-05 |
| $\beta_1$ | 0.9 | 0.9 | 0.9 |
| $\beta_2$ | 0.999 | 0.999 | 0.999 |
| weight decay | 1E-06 | 1E-06 | 1E-06 |
| num iterations | 200k | 200k | 200k |

Table 11: Hyper-parameter setting of prompt-based SFT on pretrained GPT models.

| HP | GPT-S | GPT-M | GPT-L |
|---|---|---|---|
| optimizer | AdamW | AdamW | AdamW |
| actor_lr | 3E-07 | 3E-07 | 3E-07 |
| critic_lr | 0.0003 | 0.0003 | 0.0003 |
| $\beta_1$ | 0.9 | 0.9 | 0.9 |
| $\beta_2$ | 0.999 | 0.999 | 0.999 |
| actor_adam_eps | 1E-07 | 1E-07 | 1E-07 |
| critic_adam_eps | 1E-07 | 1E-07 | 1E-07 |
| weight decay | 0 | 0 | 0 |
| epochs | 1 | 1 | 1 |
| update timestep | 512 | 512 | 512 |
| surrogate epoch | 16 | 16 | 16 |
| surrogate batch size | 32 | 16 | 8 |
| $\beta$ | 0.1 | 0.1 | 0.1 |
| $c$ | 0.01 | 0.01 | 0.01 |
| $\epsilon_{clip}$ | 0.2 | 0.2 | 0.2 |
| $\lambda$ | 1.0 | 1.0 | 1.0 |

Table 12: Hyper-parameter setting of PPO for pretrained GPT models. $\epsilon_c lip$ is the clipping parameter $\epsilon$ shown in Equation 2. $\beta$ and $c$ are weights for KL divergence and entropy as shown in Equation 3. $\lambda$ is the coefficient for SFT loss.

## A.5 Extra Results

### A.5.1 Comparing of different control types

We provide extra results for Section 4.3 in Table 13, which includes the results on both CNNDM and NYT.

| Model | Setting | CNNDM | | | | | NYT | | | | |
|---|---|---|---|---|---|---|---|---|---|---|---|
| | | R1↑ | R2↑ | RL↑ | B.S.↑ | Error↓ | R1↑ | R2↑ | RL↑ | B.S.↑ | Error↓ |
| | Prompt | 38.14 | 15.71 | 38.91 | 62.61 | 26.13 | 44.17 | 27.15 | 40.54 | 66.50 | 36.22 |
| | Prompt+RL | 35.67 | 14.64 | 38.73 | 61.86 | 13.61 | 47.57 | 30.33 | 42.88 | 67.82 | 18.81 |
| Equal | Prompt+filter | 37.90 | 16.26 | 37.42 | 61.89 | 12.47 | 47.60 | 30.32 | 42.02 | 67.80 | 17.80 |
| | Prompt+RL+filter | 37.56 | 16.10 | 38.15 | 62.23 | **8.35** | 47.76 | 30.34 | 42.15 | 67.71 | **8.72** |
| | Prompt | 37.08 | 14.68 | 36.64 | 61.88 | 0.47 | 45.81 | 28.52 | 41.22 | 67.17 | 29.45 |
| | Prompt+RL | 37.37 | 14.83 | 36.99 | 62.10 | 0.40 | 44.83 | 28.78 | 40.82 | 66.67 | 0.99 |
| Less | Prompt+filter | 36.90 | 15.72 | 35.87 | 61.13 | **0.22** | 46.68 | 29.87 | 41.53 | 66.87 | 2.09 |
| | Prompt+RL+filter | 36.90 | 15.72 | 35.87 | 61.13 | **0.22** | 46.72 | 30.43 | 42.03 | 65.97 | **0.33** |
| | Prompt | 38.00 | 15.43 | 37.82 | 62.41 | 41.87 | 44.01 | 27.12 | 40.22 | 66.62 | 2.27 |
| | Prompt+RL | 35.75 | 14.83 | 38.88 | 61.79 | 13.85 | 42.45 | 25.94 | 39.89 | 65.85 | 1.32 |
| More | Prompt+filter | 38.53 | 16.44 | 37.64 | 62.13 | 23.05 | 47.78 | 30.63 | 42.39 | 68.00 | 1.42 |
| | Prompt+RL+filter | 37.43 | 16.26 | 37.92 | 62.22 | **6.01** | 47.77 | 30.55 | 42.30 | 68.00 | **1.02** |
| | Prompt | 36.38 | 15.03 | 38.65 | 61.96 | 5.76 | 44.78 | 27.87 | 41.13 | 67.04 | 30.49 |
| | Prompt+RL | 36.10 | 14.95 | 38.99 | 61.80 | 4.53 | 47.09 | 29.74 | 42.18 | 67.63 | 10.75 |
| Between | Prompt+filter | 38.06 | 16.43 | 37.44 | 62.07 | 1.15 | 47.13 | 29.70 | 41.37 | 67.47 | 6.76 |
| | Prompt+RL+filter | 37.85 | 16.28 | 37.45 | 62.00 | **1.09** | 48.12 | 30.57 | 42.24 | 67.77 | **3.26** |

Table 13: Comparison of four control types in the multiple-type control setting using GPT-S on NYT datasets.

### A.5.2 Comparing between actor-critic model and actor only model

Another experiment is done to check the effect of using actor-critic model in comparison with actor-only model. The details of these two settings has been discussed in Section 2.1. We conduct experiments with both settings, and consider fine-tuning GPT-small model for single-type control. The results are given in Table 14. For the case without sample filtering, the model trained with actor-critic RL perform better than the model trained with actor-only RL in terms of control accuracy on both datasets. With sample filtering, actor-critic method still significantly outperforms actor-only method on NYT, but slightly worse than actor-only method on CNNDM. On NYT, rule-based reward model achieves the lowest and second lowest in the cases with and without sample filtering respectively. Meanwhile, the trainable reward models also works well.

| Setting | NYT | | | | | CNNDM | | | | |
|---|---|---|---|---|---|---|---|---|---|---|
| | R1↑ | R2↑ | RL↑ | B.S.↑ | Error↓ | R1↑ | R2↑ | RL↑ | B.S.↑ | Error↓ |
| Prompt | 47.37 | 29.22 | 42.25 | 67.70 | 13.46 | 37.45 | 15.24 | 37.62 | 62.31 | 11.89 |
| Prompt+RL+Rule (A-C) | 47.66 | 29.49 | 42.70 | 67.97 | 12.77 | 37.31 | 14.94 | 38.92 | 61.83 | 7.39 |
| Prompt+RL+Rule (A) | 47.64 | 29.53 | 42.04 | 67.96 | 12.94 | 37.74 | 15.57 | 38.20 | 62.27 | 10.98 |
| Prompt+Filter | 48.35 | 30.77 | 42.67 | 67.91 | 10.28 | 38.26 | 16.06 | 37.39 | 61.94 | 10.48 |
| Prompt+RL+Filter (A-C) | 48.31 | 30.94 | 42.82 | 67.98 | **9.55** | 37.34 | 15.71 | 38.75 | 61.22 | **6.29** |
| Prompt+RL+Filter (A) | 47.76 | 30.08 | 42.07 | 67.59 | 9.70 | 38.66 | 16.64 | 38.55 | 62.10 | 9.61 |

Table 14: The comparison of control performance of GPT-S for single-type control ("equal to") after fine-tuning by RL with and without critic models.

### A.5.3 Effect of SFT loss

As was discussed in Section 3.4, the actor loss involves a term of SFT loss, which is controlled by $\lambda$. We conduct an extra experiment on CNNDM by comparing the tuned GPT-S models using different $\lambda$s for both the case of single and multiple control types. The results are given in Table 15,

which shows that a suitable $\lambda$ is helpful in perserving the performance on downstream task, and the control accuracy will not be largely affected in most cases. Also, the optimal value of $\lambda$ differs in the cases of SG and MU, thus hyper-parameter tuning is usually needed.

| $\lambda$ | SG | | | | | MU | | | | |
|---|---|---|---|---|---|---|---|---|---|---|
| | R1 | R2 | RL | B.S. | Error↓ | R1 | R2 | RL | B.S. | Error↓ |
| 0.01 | 36.87 | 15.17 | 37.23 | 62.10 | 8.93 | 37.28 | 15.42 | 38.55 | 62.18 | 15.16 |
| 0.03 | 36.69 | 14.83 | 37.06 | 61.89 | 8.93 | 37.81 | 15.95 | 38.94 | 62.39 | 18.04 |
| 0.1 | 37.36 | 15.20 | 37.35 | 62.29 | 8.54 | 36.85 | 15.24 | 37.99 | 61.78 | 14.38 |
| 0.3 | 37.87 | 15.52 | 37.92 | 62.44 | 7.97 | 36.54 | 15.07 | 37.76 | 61.69 | 14.55 |
| 1 | 37.92 | 15.83 | 37.57 | 62.26 | 7.78 | 37.06 | 15.26 | 38.00 | 61.92 | 14.57 |
| 3 | 38.09 | 15.96 | 37.71 | 62.29 | 7.95 | 37.09 | 15.36 | 37.78 | 61.94 | 15.16 |

Table 15: The effect of SFT loss. $\lambda$ is the hyper-parameter discussed in Section 3.4.

### A.5.4 COMPARING WITH TRAINABLE REWARD MODELS.

The **trainable reward models** can be either BERT or GPT-style models, which are trained to score the generated text by concatenating it with the SCP (or the user utterance with control instructions) as input. We merge a set of randomly sampled (standard) control prompt and simulated generation with random lengths between 50 and 150 to formulate the simulated input. Then we apply the formula in Table 1 to get the labelled reward. We build a simulated dataset with 100,000 examples using the original CNNDM and NYT datasets. Then we fine-tune the pre-trained GPT-medium or BERT-large (from Huggingface) as the trainable reward models to predict reward labels. In terms of the trainable reward model performance on scoring the simulated output sequences, fine-tuned GPT-medium gives a test MSE of normalized length ($L_{target}/L_{max}$, where $L_{max}$ is the maximum output length 1024 of the used LLM) about $2e-4$ while BERT-large is around $1.5e-3$, which are significantly worse than rule-based reward model with a scoring MSE of 0. Note that although the trainable reward models use rule-based labels as in Table 1, they do not require standard prompts for calculating reward scores. Therefore, they may have better generalization than rule-based method, especially for out-of-domain user expression. The results are given in Table 16, which includes

| Setting | NYT | | | | | CNNDM | | | | |
|---|---|---|---|---|---|---|---|---|---|---|
| | R1↑ | R2↑ | RL↑ | B.S.↑ | Error↓ | R1↑ | R2↑ | RL↑ | B.S.↑ | Error↓ |
| Prompt | 47.37 | 29.22 | 42.25 | 67.70 | 13.46 | 37.45 | 15.24 | 37.62 | 62.31 | 11.89 |
| Prompt+RL+Rule | 47.66 | 29.49 | 42.70 | 67.97 | **12.77** | 37.31 | 14.94 | 38.92 | 61.83 | **7.39** |
| Prompt+RL+GPT | 47.65 | 29.52 | 42.62 | 68.07 | 12.80 | 37.67 | 14.87 | 38.24 | 62.01 | 9.26 |
| Prompt+RL+BERT | 46.79 | 28.78 | 41.80 | 73.16 | 13.23 | 37.44 | 15.05 | 38.35 | 70.16 | 9.12 |
| Prompt+filter | 48.35 | 30.77 | 42.67 | 67.91 | 10.28 | 38.26 | 16.06 | 37.39 | 61.94 | 10.48 |
| Prompt+RL+Rule+filter | 48.31 | 30.94 | 42.82 | 67.98 | **9.55** | 37.34 | 15.71 | 38.75 | 61.22 | **6.29** |
| Prompt+RL+GPT+filter | 48.56 | 30.89 | 43.07 | 68.01 | 11.01 | 37.13 | 15.67 | 37.88 | 61.51 | 8.81 |
| Prompt+RL+BERT+filter | 48.42 | 30.75 | 42.92 | 73.53 | 10.94 | 36.61 | 15.09 | 37.01 | 69.17 | 7.83 |

Table 16: Results by using different reward models for length control with single control type ("equal to") and GPT-M, where we consider the case of both with and without sample filtering.

the results on both CNNDM and NYT. Overall, both BERT or GPT models can achieve similar performance in serving as a trainable reward model. Compared to rule-based model, the advantage of BERT or GPT based reward models include a potentially better generalization ability for out-of-domain utterances and no requirement of SCPs.

### A.6 LEARNING CURVES OF STANDARD PROMPT EXTRACTION

We provide the learning curves of two types of SPE in Figure 5. For GPT-based extractor, the accuracy is 1 only if the generated SCP exactly matches the label. For BERT-based extractor, we

calculate the validation accuracy on a case-by-case basis: If the ground truth SCP type is "none", the accuracy is always 1; if the ground truth SCP type is "more than", we only match the minimum value and check if the minimum value is smaller than maximum value; if the ground truth SCP type is "less than", we only match the maximum value and check if the minimum value is smaller than maximum value; if the ground truth SCP type is "equal to" or "between", we match both of minimum and maximum values. As is shown in Figure 3, both of the SPEs converge well with a validation proportion of matching rate close to 100% in later validation steps. Meanwhile, we find the both BERT and GPT-based extractors performs fairly well on out-of-sample augmented prompts, which demonstrates strong generalization ability to new control prompts. For BERT-base, the validation curve and accuracy curve of model on out-of-sample augmented prompts converge slower than in-sample augmented prompts with a right-shift, but the accuracy values in later steps can even surpass that of in-sample validation curve. Notes than we only fine-tune the pre-trained GPT-small and BERT-base from Huggingface, which indicates the noise introduced by the extractors can generally be neglected in practice with same or larger size models.

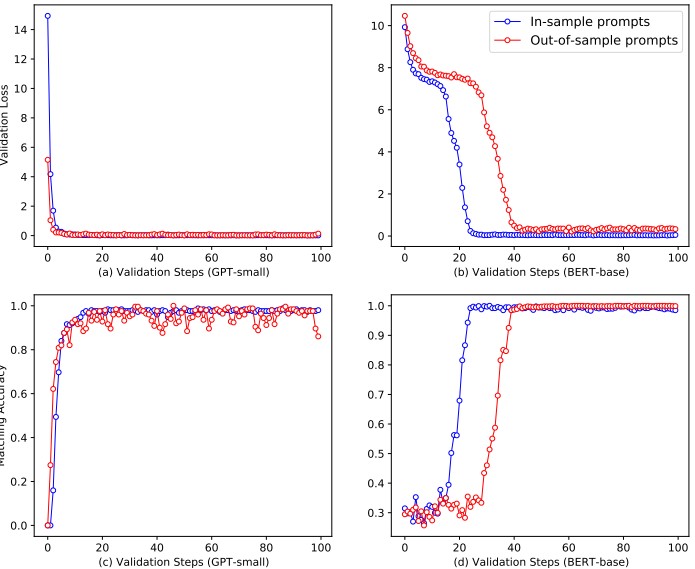

Figure 3: Learning Curves of Standard Prompt Extractors. (a) Validation losses of GPT extractor. (b) Validation losses of BERT extractor. (c) Matching accuracy of GPT extractor. (c) Matching accuracy of BERT extractor. We show the curves of validation cross entropy and matching rate for both cases.

## A.7 LEARNING CURVES OF REINFORCEMENT FINE-TUNING

To analyze the learning behavior, we visualize the learning curves of the policy loss and value loss on training set, reward (negative error normalized by the maximum length of 1024) and BERTscore (F1, in proportion) on validation set for a range of validation step. The results are generated by small GPT-2 model on both NYT and CNNDM for single-type control (with only one control instruction which is "equal to"), which are shown in Figure 4. We can see that as the decrease of policy loss and value loss, the validation reward increases relatively smoothly, while there is no clear decreasing trend of validation BERT score. The indicates that even with small GPT-2 model, the relevance can be preserved as the control accuracy increase during the RL finetuning.

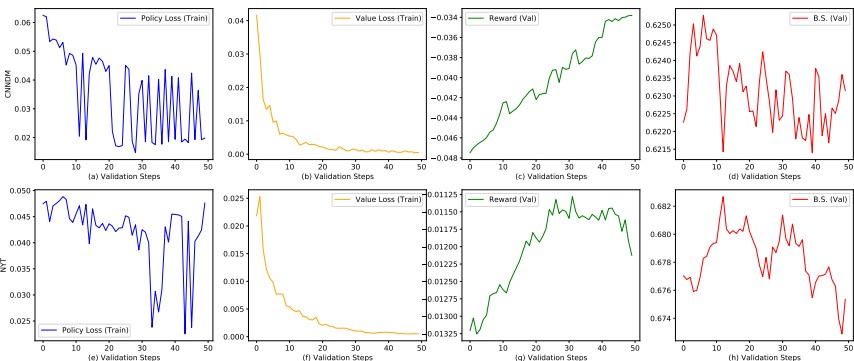

Figure 4: The Diagram of Learning Curves with GPT-S for single-type control instruction (only for "equal to") without sample filtering..

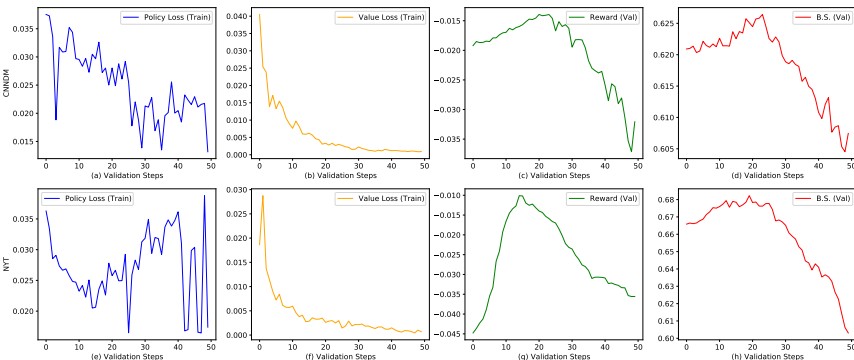

Figure 5: The Diagram of Learning Curves with GPT-S for multi-type control instructions without sample filtering.

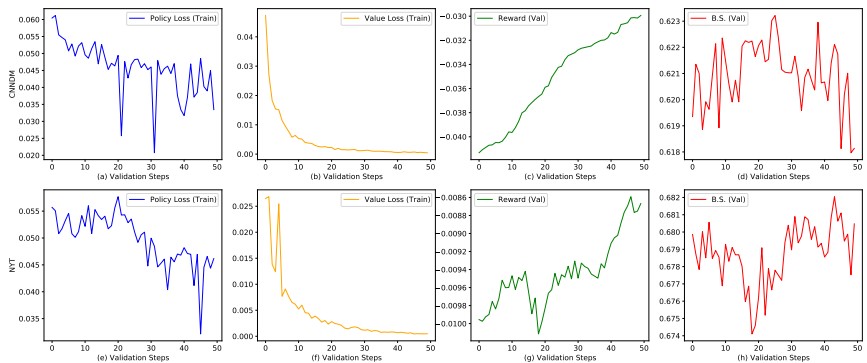

Figure 6: The Diagram of Learning Curves with GPT-S for single-type control instruction (only for "equal to") with sample filtering.

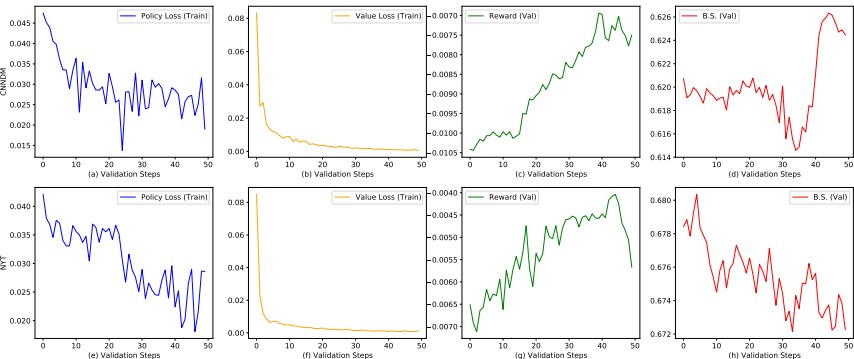

Figure 7: The Diagram of Learning Curves with GPT-S for multi-type control instructions with sample filtering.

