# OpenReview forum: "Prompt-Based Length Controlled Generation with Reinforcement Learning"
_ICLR.cc/2024/Conference — ICLR 2024 Conference Withdrawn Submission_

### Official Review · Reviewer_Dv7r · 2023-10-19

**Soundness:** 3 good
**Presentation:** 3 good
**Contribution:** 2 fair
**Rating:** 5
**Confidence:** 4

**Summary:**

This paper investigates length-controllable summarization. First (“+Prompt”), the proposed method controls the summary length by indicating the desired length in the prompt (similar to Fan et al., 2018). Second (“+RL”), to add/enhance the length-controllable capability of summarization models, the paper applies the RL method (PPO algorithm) to fine-tune GPT models using a rule-based reward. The reward simply compares the length of the generated text against the desired length, which is specified in the input prompt, and the desired length is extracted from the input prompt using a BERT/GPT-based model. Third (“+Filter”), at the inference stage, multiple summaries are sampled, and the output is the one that yields the highest reward.

The experiments were conducted on CNN/DailyMail and NYT which are standard news summarization datasets. The paper selected three sizes of GPT models (124M, 355M, 774M) as the backbone and fine-tuned these models using their proposed methods. The prompt templates were manually crafted covering many standard length-control prompts. The results show improvements over the standard prompting method (similar to Fan et al., 2018) in terms of achieving the target length while maintaining ROUGE scores.

**Strengths:**

1) The paper shows improvement in length-control ability while maintaining the ROUGE scores.

2) The paper proposes and investigates different prompt extractors, and shows that a BERT-based model achieves perfect accuracy in both seen and unseen prompts.

3) The paper is the first (or one of the first) to apply the PPO algorithm to length-controllability in summarization.

4) The ablation study shows that a simple rule-based reward performs better than model-based rewards.

**Weaknesses:**

1) The main contributions of this paper are very incremental. For example,

    - 1.1) Controlling the length by input prompts has already been done by (Fan et al., 2018) and CTRLsum (He et al., 2022).
    - 1.2) Applying RL to controlling the length has already been done by (Bian et al., 2019)
    - 1.3) Sample filtering can be considered (I believe) as a weaker version of minimum risk decoding e.g., Freitag et al., 2022
    - 1.4) The relevant references  CTRLsum (He et al., 2022) and  (Bian et al., 2019) are missing in the paper

2) The paper mentions LLMs (e.g., GPT-4, LLaMA, etc.) which are much larger and more capable than the baseline selected in this work (GPT). So, I’m not sure if the findings in this paper would transfer to those larger models (with emergence properties). I believe these larger models are becoming more accessible to researchers now, so I’m quite surprised about the model choice in this paper. Also, there are other more commonly used models such as BART, T5, and Pegasus which have fine-tuned weights on summarization tasks.

3) This paper doesn’t compare against any existing methods. The authors list some existing approaches in Section 2.2; however, in the experiments, none of them are compared against.

Note that the weaknesses #2 and #3 are minor compared to weakness #1.

References:
- (He et al., 2022) CTRLsum: Towards Generic Controllable Text Summarization
- (Fan et al., 2018) Controllable Abstractive Summarization
- (Bian et al., 2019) Controllable length control neural encoder-decoder via reinforcement learning
- (Freitag et al., 2022) High Quality Rather than High Model Probability: Minimum Bayes Risk Decoding with Neural Metrics

**Questions:**

What are your thoughts regarding the weaknesses? How do you think this approach could be applicable in the era of large language models (with stronger emergent abilities such as length control)?

---

> ### Author Response · Authors · 2023-11-22
>
> Thanks for your comments and suggestions. We provide discussions on the novelty, model size and comparison with other methods in General Response to Reviewers as these are concerned by all reviewers.
>
> In fact, the whole version of proposed method can be applied in LLMs. SPEs and sample filtering are definitely useful for LLMs in improving the length control accuracy. The effectiveness of RL depend on the length control ability of base model before using it. If the base model already has very high length control accuracy, the proportion of improvement will be relatively small.

---

### Official Review · Reviewer_JNfU · 2023-10-31

**Soundness:** 2 fair
**Presentation:** 2 fair
**Contribution:** 2 fair
**Rating:** 3
**Confidence:** 4

**Summary:**

This paper aims to control the length of generated summaries from a language model. The authors approach the goal by a prompt-based method that uses a rule-based reward function and PPO fine-tuning. The experiments conducted on the GPTs (with 124M, 355M, and 774M parameters) demonstrate that the proposed method can fine-tune the LM to be more able to be controlled the length through prompt.

**Strengths:**

* This paper proposes a reasonable method to control an LM to generate response with a length condition. This method is simple and can be effective. Specifically, this method mainly adopts PPO to optimize the LM with the authors’ designed rewards. The authors propose two variations as the reward function: (1) A standard prompt extractor (SPE) plus a rule-based reward function (Table1); (2) A GPT2/BERT-based trained reward model. Both variations use the synthetic, designed standard control prompts (SCP) to train the SPE or the reward model. The authors also propose to use the above reward functions to further select the generated summaries in the end.
* The experiments contain multiple quantitative analyses for reference. They already include the comparison among different control types, and out-of-domain length condition prompt templates.
* Most parts of the paper are clear.

**Weaknesses:**

* While this paper puts emphasis on LLM, the experiments use models with 124M, 355M and 774M, which can be controversial to be claimed as LLM. The behavior of an LM can be significantly different when the size is in Million and Billion scales. Also, which GPT is used as the main model is not specified. Because the paper only mentions the word “GPT”, I will guess it is the GPT1 (Radford et al., 2018) or the GPT2 used for the SPE.
  * Radford, Alec, et al. "Improving language understanding by generative pre-training." (2018).
* Novelty, or writing issue: Subsection 3.4 turns out to be an introduction to PPO instead of a proposed method. The added KLD penalty is also a variation proposed in (Schulman et al., 2017) and similar kinds of KLD penalty has been also added to PPO in prior work, such as (Ziegler et al., 2019). The authors can consider reorganizing the section.
  * Ziegler, Daniel M., et al. "Fine-tuning language models from human preferences." arXiv preprint arXiv:1909.08593 (2019).
* Technical issue: The definition of the advantage function is not conventional here. Specifically, A is often defined as Q(s,a)-V(s) or r + \gamma V(s’) - V(s). But in Section 3.4, the authors say the A is r - V(s,a). First, V is usually used for the state value function, whose input will only have the state. If the input includes both state and action, it is often said to be the state-action (Q) value function. Therefore, I'm wondering that is the V(s,a) should be V(s) here or the advantage used in the experiment is actually r - Q(s,a)?
* The experiments can have one baseline that does NOT use length condition prompts to fine-tune the model. This baseline can help readers understand how an “original” setup model performs on the test sets.
* The experiments miss some important details. I have checked the appendices but haven’t found them.
  * How many samples are generated for the sample filtering?
  * What is the used sampling method in the inference stage, including the hyperparameters?
* More discussion needed:
  * How do the authors view that RL+filter (BERT) receives the best BERTScore in Table 7 and 16?
  * What do the generated examples look like?
  * What kind of errors can happen? Is there a case that the summary is within the given length but the summary is actually not complete?

**Questions:**

* Some typos examples:
  * In Introduction:  “It is expensive to use human for labelling…” → labeling
  * In Section 3.2: “The Appendix A.5.3” should be A.5.4 in the manuscript.
  * In Section 4.4: “THe results are give in Tabel 6” → given

---

> ### Author Response · Authors · 2023-11-22
>
> Thanks for your comments and suggestions, which should be very helpful for improving our paper.
>
> **Q: Which GPT is used as the main model is not specified.**
>
> A: In fact, we apply GPT-2 architectures in our experiment. The three pretrained model is downloaded from Huggingface with card names "gpt2", "gpt2-medium" and "gpt2-large".
>
> **Q: Technical issue: The definition of the advantage function is not conventional here.**
>
> A: In the context of length controlled text generation, the input prompt is generally not associated with the goodness of the generated sequence (in terms of control accuracy). Meanwhile, the reward cannot be computed until the end of generation. Thus, the state value is not clearly defined. In this case, we consider the estimated advantages as the reward of actions in the current step minus the expected Q value of the last-step action by the critic network. This modified setting guarantees that the advantage is associated with the goodness of actions from current policy w.r.t. expected Q values from the old policy. Thus, the policy can be effectively optimized. Experiment show that this setting works well in both convergence and stability.
>
> **Q: How many samples are generated for the sample filtering?**
>
> A: We generate 8 samples for sample filtering in our experiment. This information is mentioned as the number of output sequence in Section 4.2.1.
>
> **Q: What is the used sampling method in the inference stage, including the hyper-parameters?**
>
> A: We apply the generate function in Pytorch transformer library, and use beam-search decoding. The number of beams is set to be 16 and the number of beam group is set to be 2, max_length is set to be 768. Other hyper-parameters are set to be the default ones of the function.

---

### Official Review · Reviewer_8j8F · 2023-10-31

**Soundness:** 2 fair
**Presentation:** 3 good
**Contribution:** 2 fair
**Rating:** 5
**Confidence:** 4

**Summary:**

In this work, for length controlled generation, The authors introduce a prompt extractor to obtain a standard control prompt, which contains metadata for controlling the length, from arbitrary user input. They define a set of standard length control types along with their corresponding rule-based reward functions. The pretrained LMs are finetuned to output while considering the standard control prompt through a modified PPO using the specified reward functions. Experimental results demonstrate an enhanced control accuracy while preserving the ability to perform downstream tasks in two summarization tasks.

**Strengths:**

- The proposed method is simple and efficient to control output length of LMs.
- The paper clearly defines a set of standard control types with appropriate reward functions.
- The paper is well written and easy to follow.

**Weaknesses:**

- It appears that there is a significant improvement in the control settings of 'Equal' and 'Between' when considering the core setting between `Prompt` and `Prompt + RL`. However, it remains unclear whether the improvement persists when the method is integrated into larger LMs such as LLaMA. This limits the extent of their contributions, despite the potential practical applicability of the method due to its simplicity.
- The paper does not compare to existing methods, such as LenAtten and LAAM, which could be adapted to the pretrained LMs used in this paper. While I understand some parts of these methods might not directly apply to this study,  at least the control target of "equal to" a specific length should be compared.
- The paper exclusively concentrates on the length control ability, rather than enhancing the downstream tasks. It would be beneficial if the reward functions for controllability and preference reward models, such as [1], were combined to enhance both the length control ability and summarization performance simultaneously. Moreover, including human evaluation for the generated summary would be valuable.

    [1] Learning to summarize from human feedback

- The paper lacks a comparison between the modified PPO and the standard PPO.

**Questions:**

- see weaknesses
- minor comments
    - What N of sample filtering is used?
    - What is SG and MU in Table. 6?
    - The table caption should be positioned at the top of the table.

---

> ### Author Response · Authors · 2023-11-22
>
> Thanks for your comments and suggestions. We would like to answer some of the questions as follows:
>
> **Q:What N of sample filtering is used?**
>
> A: We generate 8 samples for sample filtering in our experiment. This information is mentioned as the number of output sequence in Section 4.2.1.
>
> **Q: What is SG and MU in Table. 6?**
>
> A: “SG” refers to solely involving the control type of “Equal to”. “MU” refers to involving all the four control types with equal proportion in augmented input. This information is also mentioned in Section 4.2.1.

---

### Official Review · Reviewer_ZCo7 · 2023-10-31

**Soundness:** 3 good
**Presentation:** 3 good
**Contribution:** 2 fair
**Rating:** 5
**Confidence:** 4

**Summary:**

The goal of this work is training models that can accept natural-language length constraints as part of the prompt, including "equals", "less/greater than" and "between" styles of constraint. The work uses existing task data, particularly CNNDM and NYT summarization data. They use hand-crafted prompts to add natural language constraints with various surface forms to this existing data, and train models to convert these constraints into a structured format that can be evaluated automatically. They compare various types of training including both finetuning and RL-based methods for encouraging models to follow the given length constraints.

**Strengths:**

- Rule based rewards seem effective and more natural than 0-1 error that is often used for constraints
- the standard prompt extractors (SPE) seem to achieve a high accuracy, including on held-out prompt templates
- Adding RL + filter seems to improve the ability of models to adhere to length constraints

**Weaknesses:**

- Much of the paper is focused on using existing techniques, e.g. training models on the length of existing texts (this was used for length-controlled T5 infilling for instance) and PPO RL
- While the SPE accuracy seems high on held-out templates, all templates were written by the authors and are unlikely to cover the diversity of what humans might use in the wild. It would be useful to find a way to test generalizability to real user inputs. This is particularly important because the authors specifically frame this aspect of the paper as handling diverse inputs, and so truly demonstrating that this component of the pipeline (which is a significant part of the contribution) indeed generalizes to diverse surface forms. Otherwise, it is not completely clear why the authors would not just define a standard format, as the input prompts are all defined by the authors anyway.
- While RL does seem to result in lower constraint error, it also (by inspection, e.g. in table 3) seem to often lower the automatic quality metrics. It would also be useful for the authors to include bold and underline values in all columns, not just constraint error.

**Questions:**

Please correct me if there is anything I missed in terms of contributions

---

> ### Author Response · Authors · 2023-11-22
>
> Thanks for your comments and suggestions. We add a discussion of the novelty in General Response to Reviewers.
>
> As to the generalization experiment of SPE for unseen prompts, we build prompt template data by ourselves. We believe it will be better if we can use public dataset or data from the users in the real world. However, we do not find public datasets for length control templates.  Also, it seems impractical to get diverse user data as these data are only available to firms that provide LLM services to public users.

---

### Author Response · Authors · 2023-11-22
**General Response to Reviewers**

We sincerely thanks all reviewers for their valuable comments and suggestions. Here we response to some of common concerns.

**Novelty:**

Although some major components of our methods have been studied by existing works, it is still important to develop systematical length control technique for GPT-type models in the era of LLMs. The designing should be carefully considered in this context. Moreover, as we know, all existing length control methods focus on the case of “Equal”, but in practice the users can have more diverse requirements of length control. Our method can handle diverse control information for different control types in both RL and sample filtering. The SPEs is novel for extracting length control information with both BERT and GPT architectures, which is shown to be effective with strong generalization ability. (Note that we do not find public datasets for length control templates, and it seems impractical to get diverse user data for this.)

**Model size:**

We use pre-trained GPT-2 models with three different sizes and focus on the case of text summarization, which is a scenario that is in most demand of length controlled generation. Since this particular task does not require very large LLM to achieve satisfactory results, we did not use LLMs with large-scale multitask SFT. In fact, all the components in the proposed method can be directly applied for larger LLMs. We believe that compared with smaller models, larger models can have stronger power in learning the length control ability without losing the power for downstream tasks. By using LoRA tuning, LLMs can learn the length control ability without changing the parameters of the original model for downstream task.

**Comparison with other methods:**

We have shown the significant improvement of our method over baseline prompt-based length control method that is widely used in LLMs. The existing length control methods such as LAAM and LenAtten are not fully open-source. One need to write or modify many parts of the code, re-pertaining and retuning on many hyper-parameters to apply them in large-size GPT-type models. Also, these two methods can only be used for one control type (Equal) studied in our study, while the other three control types are highly in demand by users of real applications. Furthermore, our method can be applied along with these two existing methods to achieve further improvement, thus it is a powerful tool for building models with strong length control ability.